# DiffGANPaint: Fast Inpainting Using Denoising Diffusion GANs

**Moein Heidari**
Iran University of Science and Technology
moeinheidari7829@gmail.com

**Alireza Morsali**
McGill University
alireza.morsali@mail.mcgill.ca

**Tohid Abedini**
Iran University of Science and Technology
t_abedini@comp.iust.ac.ir

**Samin Heydarian**
Iran University of Science and Technology
samin_heydarian@comp.iust.ac.ir

## Abstract

Free-form image inpainting is the task of reconstructing parts of an image specified by an arbitrary binary mask. In this task, it is typically desired to generalize model capabilities to unseen mask types, rather than learning certain mask distributions. Capitalizing on the advances in diffusion models, in this paper, we propose a Denoising Diffusion Probabilistic Model (DDPM) based model capable of filling missing pixels fast as it models the backward diffusion process using the generator of a generative adversarial network (GAN) network to reduce sampling cost in diffusion models. Experiments on general-purpose image inpainting datasets verify that our approach performs superior or on par with most contemporary works.

## 1 Introduction

Image Inpainting is the task of reconstructing missing regions in an image. As an inpainting approach requires strong generative capabilities, most of the contemporary works rely on GANs (Zheng et al., 2022) or Autoregressive Modeling (Yu et al., 2021). Capitalizing on the advances in diffusion models, a different line of research is the DDPM-based image synthesis (Meng et al., 2021; Lugmayr et al., 2022). Despite their impressive results, DDPM-based models suffer from computationally expensive sampling procedures. To circumvent this, we propose DiffGANPaint, an inpainting method that leverages trained DDPM and uses a trained GAN model during the reverse process to generate the inpainted image. Thus, our model is a mask-agnostic approach that allows the network to generalize to any arbitrary mask during inference using the generation capabilities of DDPMs. Experiments on the diverse datasets demonstrate generalization in inpainting semantically meaningful regions.

## 2 Related Work

Most Existing literature on inpainting methods follow a standard configuration and use diverse GAN-based structures (Cha & Kim, 2022). Despite remarkable image synthesis performance, in these methods, still, pixel artefacts or colour inconsistency occur in synthesized images during the generation process. In a different direction, (Lugmayr et al., 2022) use image prior and a pre-trained Denoising Diffusion Probabilistic Model for generic inpainting. Similar to this, we propose a novel method which uses a trained GAN in the reverse diffusion process to ameliorate the rapidity and sample quality performance (Elaborated in section 3).

## 3 Methodology

Our approach (see Figure 1) is comprised of a diffusion model that denoises an image using the diffusion process, which is then used to prepare the image for inpainting using the generator of

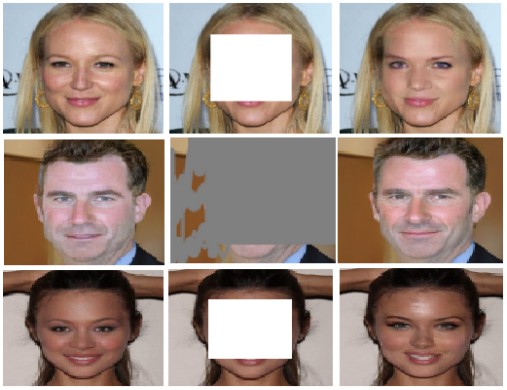

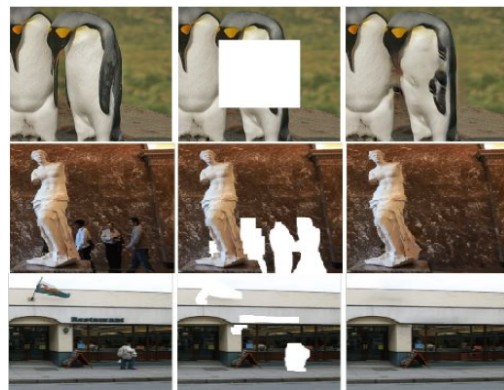

Figure 2: Visual examples of DiffGANPaint results on CelebA-HQ faces. From left to right, shows the original image, input masked image and result image.

Figure 3: Visual examples of DiffGANPaint results on generic images. From left to right, shows the original image, input masked image and result image.

a trained GAN model that generates the image. Specifically, the inpainting is performed by first denoising the input image using the diffusion process, then extracting the masked region from the original image, and finally inpainting the masked region using the GAN generator. Therefore, we can concurrently leverage the structure consistency attained by DDPM, and high-quality rapid samples achieved by the GAN generator. Our approach is illustrated more in Figure 1.

## 4 EXPERIMENTS AND RESULTS

We perform experiments for inpainting tasks on generic data and the CelebA-HQ faces datasets [1]. We use the trained guided diffusion and GAN model on Imagenet (Dhariwal & Nichol, 2021). The visual results of the generation process are provided in Figure 2 and 3. As shown, DiffGANPaint produces higher visual quality with a low computational budget. Concretely, our approach can produce samples in fewer steps while trading off the sample quality.

```python
timesteps=100
def denoise_diffusion(x, model):
    noise = torch.randn_like(x)
    for i in range(timesteps):
        eps = torch.randn_like(x)
        x = x + torch.sqrt(torch.tensor(2.0)) * eps
        model_input = torch.cat([x, noise])
        out = model(model_input)
        x = x + out * np.sqrt(1/timesteps)
    return x

def test_inpainting(img, gan,mask):
    denoised_img = denoise_diffusion(img,gan)
    model_input = torch.cat([denoised_img, mask
                                            ])
    output = gan(model_input)
    output = output * mask + masked_image * (1
                                        - mask)
```

## 5 CONCLUSION

This paper leverages the benefits of a novel denoising diffusion probabilistic model and GAN model solution for the image inpainting task. Specifically, we exploit a trained diffusion

Figure 1: **DiffGANPaint Inpainting Procedure in Pytorch Style:** Pseudo code to generate the inpainted image using the corresponding mask, trained diffusion model and generator.

model and modify the reverse diffusion using a GAN generator to paint images with better mode coverage and sample diversity. As showcased on various datasets, our model demonstrates strong visual capabilities at a low computational cost.

---

[1]The code is available at this URL: https://github.com/moeinheidari/DIFFGANPAINT

URM STATEMENT

The first author of this paper, namely Moein Heidari, meets the URM criteria of the ICLR 2023 Tiny Papers Track. He is 23 years old outside the range of 30-50 years.

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
