# OpenReview forum: "DiffGANPaint: Fast Inpainting Using Denoising Diffusion GANs"
_ICLR.cc/2023/TinyPapers — Submitted to Tiny Papers @ ICLR 2023_

### Official Review · Reviewer_Ec6j · 2023-03-20

**Confidence:** 4

**Summary Of Contributions:**

This work proposes a method for fast in-painting by combining the diffusion models and generative adversarial methods. DDPMs were leveraged for their structured consistency and GANs were used for their fast generations.

**Rating:**

Needs Clarification (NC): a submission which does not meet the reviewing criteria and needs clarification for its described problem or solution

**Strengths And Weaknesses:**

Strengths:

1. The paper is relatively easy to follow and the idea presented is interesting.
2. The motivation was also clear, but not the application of the method.
3. The pseudocode is provided and if accepted, the published repository should help with reproducibility.
4. The paper adheres to the formatting requirements

Weaknesses:

1. Despite the interesting idea, the motivation and the advantages of the method for a realistic application remains vague.
2. The claims were not very well justified as the results were more qualitative and were not compared to any benchmarks



**Suggested Changes:**

1. Additional description/discussion on the motivation of the application would significantly strengthen the paper. I understand that the authors wanted to combine the high quality generation of diffusion models with fast generation of GANs. However, based on the pseudocode, it seemed like the Diffusion Model is first used to generate an image, then a mask is applied onto the image before a GAN is used to in-paint the mask. If that is the case, how is the computation saved since a diffusion model is used followed by a GAN? It is not clear if the initial input is an image with a mask or is this a pure generation problem? If it is a pure generation problem, then the authors could justify why is there a need for in-painting. If the initial image is a masked image, why not just directly use a Diffusion model to inpaint the mask?

2. The authors could also improve the paper by comparing their results with some benchmark, whether in terms of generation quality and/or generating timings. It is difficult for a reader to justify the authors claims that their method in-paints with a higher visual quality without comparison to at least a simple vanilla GAN or Diffusion Model.

---

### Meta-Review · Area_Chair_dtEZ · 2023-04-08

**Recommendation:** Invite to archive
**Confidence:** 4

**Metareview:**

Based on the review, the paper proposes a method for fast in-painting by combining diffusion models and generative adversarial methods. The strengths of the paper are that it is easy to follow, the idea presented is interesting, the motivation is clear, and the pseudocode is provided. The weaknesses are that the advantages of the method for a realistic application remain vague, and the claims were not well justified as the results were more qualitative and were not compared to any benchmarks.

The main message of the paper is that the proposed method combines the advantages of diffusion models and generative adversarial methods to enable fast in-painting. However, the paper lacks clarity regarding the advantages of the method for a realistic application and lacks sufficient justification for its claims.


**Summary:**

This work proposes a method for fast in-painting by combining the diffusion models and generative adversarial methods.

**Reason For Not Giving A Higher Recommendation:**

 It can be further improved when incorporating the suggested changes by the reviewer.


**Reason For Not Giving A Lower Recommendation:**

The paper has some merits as discussed by the reviewer.

---

### Decision · Program_Chairs · 2023-04-10

Invite to archive

---

> ### Comment · Area_Chair_dtEZ · 2023-06-06
> **Meet the threshold for archival**
>
> This work meets the threshold for archival, contents the URM statement and is deanonymized.